# *Annona muricata* L.-Derived Polysaccharides as a Potential Adjuvant to a Dendritic Cell-Based Vaccine in a Thymoma-Bearing Model

**DOI:** 10.3390/nu12061602

**Published:** 2020-05-29

**Authors:** Woo Sik Kim, Jeong Moo Han, Ha-Yeon Song, Eui-Hong Byun, Seung-Taik Lim, Eui-Baek Byun

**Affiliations:** 1Advanced Radiation Technology Institute, Korea Atomic Energy Research Institute, Jeongeup 580-185, Korea; kws6144@kaeri.re.kr (W.S.K.); jmhahn@kaeri.re.kr (J.M.H.); songhy@kaeri.re.kr (H.-Y.S.); 2Department of Biotechnology, College of Life Science and Biotechnology, Korea University, Seoul 136-701, Korea; limst@korea.ac.kr; 3Department of Food Science and Technology, Kongju National University, Yesan 340-800, Korea; ehbyun80@kongju.ac.kr

**Keywords:** dendritic cells, vaccine adjuvant, *Annona muricata* L., polysaccharide, Th1, cytotoxic T lymphocyte, multifunctional T cells

## Abstract

Dendritic cells (DCs) are powerful antigen-presenting cells that are often used to evaluate adjuvants, particularly for adjuvant selection for various vaccines. Here, polysaccharides (named ALP) isolated from leaves of *Annona muricata* L., which are used in traditional medicine such as for bacterial infections and inflammatory diseases, were evaluated as an adjuvant candidate that can induce anti-tumor activity. We first confirmed the phenotypic (surface molecules, cytokines, antigen uptake, and antigen-presenting ability) and functional alterations (T cell proliferation/activation) of DCs in vitro. We also confirmed the adjuvant effect by evaluating anti-tumor activity and immunity using an ALP-treated DC-immunized mouse model. ALP functionally induced DC maturation by up-regulating the secretion of Th1-polarizing pro-inflammatory cytokines, the expression of surface molecules, and antigen-presenting ability. ALP triggered DC maturation, which is dependent on the activation of the MAPK and NF-κB signaling pathways. ALP-activated DCs showed an ample capacity to differentiate naive T cells to Th1 and activated CD8^+^ T cells effectively. The systemic administration of DCs that pulse ALP and ovalbumin peptides strongly increased cytotoxic T lymphocyte (CTL) activity (by 9.5% compared to that in the control vaccine groups), the generation of CD107a-producing multifunctional T cells, and Th1-mediated humoral immunity, and caused a significant reduction (increased protection by 29% over that in control vaccine groups) in tumor growth. ALP, which triggers the Th1 and CTL response, provides a basis for a new adjuvant for various vaccines.

## 1. Introduction

Vaccines are a safe and effective way to treat, mitigate and prevent various diseases such as glioblastoma, melanoma cancer, tuberculosis, and hepatitis infection, and they help the body’s immune system to recognize and combat specific harmful diseases [1,2,3]. Importantly, to provide an effective immune response against diseases, vaccines must have adequate immunogenicity to affect various immune cell types [4]. In particular, innate and adaptive immune cells can serve as an essential fulcrum in initiating the host defense response in the early and late stages of diseases [5]. However, most vaccine candidates have a reduced capacity to induce strong innate and adaptive immune responses [6,7,8]. Interestingly, these vaccines can induce a robust immune response when used with adjuvants that can enhance the immune response [8,9]. In this regard, several studies are underway to enhance and improve the immunogenicity of vaccines by using various adjuvants, and many researchers have focused on the development of an effective adjuvant as a disease-target immune enhancer [10,11]. 

Adjuvant candidates include chemically and biologically active compounds such as microbial substances, mineral salts, emulsions, pharmaceutical agents, or natural products [12,13,14]. Among these candidates, natural products extracted or isolated from plants are now attracting attention due to many advantages, such as their availability and affordability, and minimal or no side effects [13,15,16,17]. In particular, there is a growing interest in medicines of botanical origin that lack severe side effects and have proven efficacy in traditional medicine [18,19]. 

*Annona muricata*, commonly known as Graviora, Soursop, or Guabana, is a small evergreen tree with many bioactive compounds, such as vitamins, minerals, phenolic acids, and flavonoid glycosides [20,21]. Traditionally, its leaves have been used as an ethnomedicine to treat various diseases, including hypertension, malaria, asthma, and cancer [21]. Importantly, a recent study indicated that *A. muricata* leaf extracts have potential as a health-promoting ingredient to boost the innate immune system by inducing macrophage activation [20]. In addition, Gavamukulya et al. showed that *A. muricata* leaf extracts have a direct lethal effect on various cancer cells but not healthy cells [22]. Thus, given that the *A. muricata* leaf extract is characterized by its ability to enhance the immune response effectively without showing toxicity, natural products contained in *A. muricata* leaves appear to have sufficient potential as an immunostimulant adjuvant, particularly in cancer patients that require strong innate and adaptive immunity.

Previously, our group reported that the polysaccharides isolated from leaves of *A. muricata* L., consisting of several sugars (mainly galactose, glucose, and mannose), represent a novel pharmacological and therapeutic candidate for treating neurodegenerative diseases by preventing neuronal oxidative stress [23]. In this study, we confirmed the ability of the extract to stimulate anti-tumor immunity based on its immunostimulatory effects in innate and adaptive immune responses. Additionally, to clarify its potential use as an adjuvant, we showed that *A. muricata* L. leaf polysaccharide (ALP) could boost the anti-tumor effect of and protective immune response to dendritic cell (DC)-based therapeutic vaccination in a thymoma-bearing mouse model.

## 2. Materials and Methods

### 2.1. Experimental Animals and Ethics Statement

Female 6-week-old C57BL/6 and BALB/c mice were purchased from Orient Bio Inc. (Seoul, Korea). The animals were acclimated to the following controlled conditions—temperature (25 ± 2 °C), humidity (55% ± 5%), and a 12 h light/dark cycle—at the Central Animal Research Laboratory, Korea Atomic Energy Research Institute (KAERI, Jeongeup, Korea). The animals were fed a sterile commercial mouse diet and provided with water ad libitum. The mice were monitored daily, and none exhibited any illnesses or clinical symptoms during the experiments. All the animal experiments were assessed and approved by the Institutional Animal Care and Use Committee (IACUC) of the KAERI (Permit Number: KAERI-IACUC-2019-001).

### 2.2. Preparation of ALP

*A. muricata* L. leaf was purchased from the Jechcheon herbal market, in the Chungbuk province of Korea. *A. muricata* L. leaf was freeze-dried (VD-800F; Taitec, Saitama-ken, Japan) and ground into small granules. Lyophilized powder was washed (100 g/L) in deionized water (DW) at 90 °C for 2 h. Next, these samples were passed through Whatman No. 42 filter paper and incubated with 70% ethanol at 4 °C for 18 h. Then, the 70% ethanol was removed and centrifuged. Additionally, monosaccharides and other components from the harvested samples were removed by size separation using a 10–12 kDa molecular weight cut-off (MWCO, Millipore, Burlington, MA, USA). Finally, the filtered samples were lyophilized and named as ALP. The yield of ALP expressed on a dry weight basis was 16.4%.

### 2.3. Cell Lines

The ovalbumin (OVA)-expressing mouse thymoma E.G7 (a derivative of EL4 having a C57BL/6 background) cells were obtained from ATCC (Manassas, VA, USA). The E.G7 cells were used at Passage 6. The cells were grown in the presence of RPMI 1640 medium (Biowest, Nuaille, France) with 10% fetal bovine serum (FBS, Biowest) and 1% penicillin/streptomycin (P/S, GIBCO, Carlsbad, CA, USA) at 37 °C with 5% CO_2_.

### 2.4. Generation and Culture of Bone Marrow-Derived Dendritic Cells (BMDCs)

Bone marrow cells were flushed out from femurs and tibias. After red blood cell (RBC) lysis using RBC lysis buffer (Sigma-Aldrich, St. Louis, Mo, USA), the lysed cells were cultured in 100 mm Petri dishes. These cells were grown in the presence of complete RPMI (c-RPMI) media after supplementing with 10% FBS, 1% P/S, recombinant granulocyte-macrophage colony-stimulating factor (GM-CSF, 20 ng/mL, JW CreaGene, Daegu, Korea), and IL-4 (0.5 ng/mL, JW CreaGene). On Days 3 and 6, c-RPMI media were added. On Day 8, to identify the CD11c^+^ population, the cells were harvested and stained with anti-CD11c monoclonal antibody (anti-CD11c mAb, eBioscience, San Diego, CA, USA), which served as a DC phenotypic marker. 

### 2.5. Cytotoxicity Analysis

To demonstrate the cytotoxicity of ALP, 8 days after their generation and culture, BMDCs were grown in a 48-well plate (1 × 10^6^/well) and stimulated with LPS (100 ng/mL; Invivogen, San Diego, CA, USA) and ALP (10, 30, 100, and 200 μg/mL) for 18 h. After harvesting non-, LPS-, and ALP-treated DCs, the cell pellets were stained with Annexin V-FITC (diluted 1:50 in Annexin V binding buffer, BD Bioscience) for 15 min at room temperature. Next, after washing the cells by adding Annexin V binding buffer, the cells were stained with propidium iodide (PI, diluted 1:25 in Annexin V binding buffer, BD Bioscience) for 10 min at room temperature. Finally, the cells labeled with Annexin V/PI were counted using flow cytometry (FACSverse, BD Bioscience, San Jose, CA, USA), and the flow cytometry (FACS) data were analyzed with the FlowJo V10 software (BD Bioscience). 

### 2.6. Cytokine Measurement

The extracellular cytokine levels in the supernatants, which were stimulated by ALP, were confirmed using an enzyme-linked immunosorbent assay (ELISA, eBioscience) kit. All the steps of ELISA were conducted according to the manufacturer’s instructions. The quantities of immunoreactive cytokine proteins (TNF-α, IL-12p70, and IL-10) were read at 450 nm in a microplate ELISA reader (Zenyth 31004 Anthos Labtec Instruments GmbH, Salzburg, Austria).

### 2.7. Intracellular Cytokine Staining in DCs

After being stimulated with ALP in the presence of GolgiStop (0.5 μg/mL, BD Bioscience) and GolgiPlug (0.5 μg/mL, BD Bioscience) for 9 h, DCs were stained using Live/Dead Aqua (Live/Dead, BV510, Invitrogen, Carlsbad, CA, USA) and anti-CD11c mAb (PE-Cy7, eBioscience) for 15 min at room temperature. Then, the cells were added to BD Cytofix/Cytoperm solution (reagent for the fixation and permeabilization of cells) for 25 min at 4 °C. After washing the cells by adding BD Perm/Wash buffer, the cells were incubated with anti-TNF-α (APC, BD Bioscience), anti-IL-12p70 (PE, BD Bioscience), and anti-IL-10 (FITC, BD Bioscience) mAbs. After staining, the cells were counted using the FACSverse and FlowJo V10 software.

### 2.8. Cell Surface Molecule Analysis

To investigate the expression levels of surface immune-related molecules for DC maturation in ALP-treated BMDCs, we treated the cells with LPS (100 ng/mL) or ALP (10, 30, 100, and 200 μg/mL) for 18 h. The cells were harvested and stained with Live/Dead, anti-CD11c (BV421, BD Bioscience), anti-CD80 (FITC, BD Bioscience), anti-CD86 (PE, BD Bioscience), anti-major histocompatibility complex-I (MHC-I, APC, eBioscience), and anti-MHC-II (PE-Cy7, eBioscience) mAbs for 15 min at room temperature. To detect the isotype controls, we stained the cells with the matching appropriate isotype immunoglobulins, including IgG2 κ (FITC, BD Bioscience), rat IgG2a κ (PE, BD Bioscience), rat IgG2a κ (APC, BD Bioscience), and rat IgG2b κ (PE-Cy7, BD Bioscience) mAbs, which served as negative controls for the respective surface molecule mAbs. After staining, the cells were counted using FACSverse, and the FACS data were analyzed with the FlowJo V10 software.

### 2.9. Antigen Uptake Capacity

After treatment with LPS (100 ng/mL) and ALP (200 μg/mL) for 18 h, DCs were added to fluorescein-conjugated dextran (0.5 mg/mL, Sigma-Aldrich) for 30 min at 37 °C. The resulting cell pellets were then harvested and washed 2 times with cold FACS buffer (phosphate-buffered saline supplemented with 0.5% FBS and 0.1 sodium azide). Finally, the washed cells were labeled with DC surface antibody (anti-CD11c, PE-Cy7). The cells (Dextran^+^CD11c^+^ cells) were counted using FACSverse, and the FACS data were analyzed with the FlowJo V10 software.

### 2.10. Antigen-Presenting Assay

To identify the antigen-presenting ability of ALP-treated DCs, we analyzed peptide–MHC-I and MHC-II complex formation using the OVA_257–264_ and Eα peptides. The DCs were treated with LPS (100 ng/mL) or ALP (30 μg/mL) in the presence of the OVA protein (500 μg/mL, Sigma-Aldrich) for 18 h to analyze peptide–MHC-I complexes. The DCs were treated with LPS (100 ng/mL) or ALP (30 μg/mL) in the presence of Eα_44–76_ (25 μg/mL, AbFrontier, Seoul, Korea) for 18 h to investigate the peptide–MHC-II complexes. Positive controls, 5 μg/mL of OVA_257–264_ (AbFrontier) and Eα_52–68_ (AbFrontier), were also used (3 h treatment). After harvesting the cells, the resulting cell pellets were washed 2 times with cold FACS buffer and then stained with anti-CD11c (PE-Cy7), anti-25-D1.16 (eBioscience), or anti-Y-Ae Abs (eBioscience) for 15 min at room temperature. The stained cells were counted using FACSverse, and the FACS data were analyzed with the FlowJo V10 software.

### 2.11. Western Blot Analysis and Nuclear Extract Preparation

The DCs were stimulated with ALP (200 μg/mL) at different times (0, 5, 15, 30, and 60 min). For cytosolic protein analysis, the cells were lysed using protein lysis buffer (RIPA buffer, Pierce, Rockford, IL, USA). The immunoblotting of proteins isolated from cells was carried out as previously described [24]. For nuclear extract analysis, the cells were lysed using the CelLytic NuCLEAR Extraction Kit following the manufacturer’s instructions (Sigma-Aldrich). The Western blotting Abs used to detect the MAPK and NF-κB signals were purchased from Cell Signaling Technology (Boston, MA, USA). 

### 2.12. Confocal Laser Scanning Microscopy

The cells (6 × 10^4^ cells per well) were cultured on coverslips coated with 0.5 mg/mL poly-L-Lysine solution (Sigma-Aldrich) at 37 °C. After 12 h, the cells were stimulated with ALP (200 μg/mL) for 1 h and were fixed with MeOH for 10 min and blocked with PBS containing 5% bovine serum albumin (BSA; Sigma-Aldrich) and 0.3% Triton X-100 (Sigma-Aldrich) for 1 h. The cells were then cultured with anti-NF-κB mAb (Calbiochem, San Diego, CA, USA) in phosphate-buffered saline containing 1% BSA. After 2 h, the cells were stained with appropriate secondary Abs conjugated with Alexa Fluor 488 for 1 h at room temperature. Next, the cells were incubated with 4’,6-diamidino-2-phenylindole (DAPI; Pierce, Rockford, IL, USA) to stain nucleic acids for 15 min. Images were obtained using a Carl Zeiss LSM510 confocal microscope (Jena, Germany).

### 2.13. Treatment of DCs with Pharmacological Inhibitors of Signaling Pathways

All the pharmacological inhibitors of MAPKs and NF-κB used in this study were purchased from Calbiochem (San Diego, CA, USA). Dimethyl sulfoxide (0.1%, Sigma-Aldrich) was used as the solvent control for inhibitor treatment. The cells were exposed to MAPK and NF-κB inhibitors for 1 h prior to stimulation with ALP (200 μg/mL) and LPS (100 ng/mL) for 18 h. The following inhibitors were used: U0126 (ERK inhibitor; 10 μM), SP600125 (JNK inhibitor; 20 μM), SB203580 (p38 inhibitor; 20 μM), and Bay11-7082 (NF-κB inhibitor; 20 μM).

### 2.14. Allogeneic Mixed Lymphocyte Reaction (MLR)

To investigate the T cell proliferation induced by ALP-treated DCs, we separated splenic CD4^+^ and CD8^+^ T cells from BALB/c mice using Miltenyi MACS microbeads conjugated with anti-CD4 or anti-CD8 mAbs (Biotec, San Diego, CA, USA) and a MACS large separation column (Miltenyi Biotec). The isolated cells (2 × 10^6^ cells/well) were next stained with carboxyfluorescein succinimidyl ester (CFSE, 1 μM, Invitrogen) and co-cultured with each set of DCs (ALP- and LPS-treated DCs, 2 × 10^5^ cells per well). After 2 days of co-culture, the T cells were stained with anti-CD4 (Alexa488, BD Bioscience) and anti-CD8 (BV510, BD Bioscience) mAbs. The stained cells were counted using FACSverse, and the FACS data were analyzed with the FlowJo V10 software. Supernatants were collected to measure the IFN-γ (BD Bioscience), IL-2 (eBioscience), and IL-5 (eBioscience) levels using ELISA.

### 2.15. Cytotoxic T Lymphocyte Activity and Multifunctional T Cell Subset Analysis

For the in vivo cytotoxic T lymphocyte (CTL) killing assay, C57BL/6 mice (6 mice per group) were immunized with PBS, non-treated DCs (non-DC), DCs pulsed with OVA_257–264_ (5 μg/mL, OVA_257–264_-DC), or ALP (200 μg/mL)-treated DCs pulsed with OVA_257–264_ (ALP/OVA_257–264_-DC) on Days 1, 3, and 5. On Day 7 of the last immunization, single-cell suspensions from the spleens of normal syngeneic C57BL/6 mice were isolated. The isolated cells were pulsed with or without 10 μg/mL OVA_257–264_ for 45 min at 37 °C. Next, the OVA_257–264_-pulsed or non-pulsed populations were loaded with either 5 mM (CFSE^high^) or 0.5 mM (CFSE^low^) CFSE at 37 °C for 10 min. Then, 2 cell populations (CFSE^high^- and CFSE^low^-stained cells) were mixed 1:1 and injected into each immunized mouse (10^7^ cells per mouse) through the tail vein. Four hours after injection, the spleens from the recipient mice (each immunized mouse) were isolated, and single-cell suspensions were prepared before flow cytometry analysis. In the flow cytometry data analysis, the elimination of OVA-_257–264_-pulsed splenocytes (CFSE^high^) in the splenocytes of each immunized mouse indicated cytotoxic activity.

For multifunctional T cell analysis, C57BL/6 mice were immunized with PBS (G1), non-treated DCs (G2; iDCs), DCs pulsed with OVA_257–264_ and OVA_323–339_ (G3; OVAs-DC), or ALP (200 μg/mL)-treated DCs pulsed with OVA_257–264_ and OVA_323–339_ (G4; ALP/OVAs-DC) on Days 1, 3, and 5. All the OVA peptides were used at a concentration of 5 μg/mL. Two weeks after the last immunization, the splenocytes of each immunized mouse were isolated and lysed using RBC lysis buffer. Next, the cells were restimulated with OVA_257–264_ (5 μg/mL) and OVA_323–339_ (5 μg/mL) in the presence of GolgiStop (0.5 μg/mL), GolgiPlug (0.5 μg/mL), and anti-CD107a antibody (Ab) (2.5 μg/mL, BD Bioscience) for 6 h at 37 °C. The cells were then stained using Live/Dead and with anti-CD3 (APC-Cy7, eBioscience), anti-CD4 (Alexa488), and anti-CD8 (PerCp-Cy5.5, eBioscience) mouse Abs (mAbs) for 30 min at 4 °C. Then, the cells were added to BD Cytofix/Cytoperm solution for 25 min at 4 °C. After washing the cells by adding BD Perm/Wash buffer, the cells were incubated with anti-IFN-γ (PE, eBioscience), anti-TNF-α (APC, eBioscience), and anti-IL-2 (PE-Cy7, eBioscience) mAbs for 30 min at room temperature. After staining, the cells were counted using FACSverse and the FlowJo V10 software.

### 2.16. Measurement of OVA-Specific Ab

OVA-specific IgG1 (Sigma-Aldrich) and IgG2a (Southern Biotech, Birmingham, AL, USA) Abs in the serum were measured using indirect ELISA. Briefly, plates were coated with OVA_323–339_ peptide (1 μg/mL) and incubated overnight at 4 °C. The wells were washed 3 times with PBS containing 0.05% (v/v) Tween 20 (PBS/Tween) and blocked with PBS containing 5% FBS at 37 °C for 2 h. After washing 3 times with PBS, diluted sera (1:500) from each immunized mice were added, and the plates were incubated for 2 h at 37 °C and washed 3 times with PBS. Then, HRP-conjugated secondary Abs against IgG1 and IgG2a were added, and the plates were incubated for 1 h at room temperature. The reaction was developed using TMB substrate (Sigma-Aldrich). After terminating the reaction, the signal at 495 nm was detected within 20 min in a microplate ELISA reader.

### 2.17. Therapeutic Potential via E.G7 Tumor Challenge

Mice were injected with E.G7 cells (2 × 10^6^) subcutaneously into the right lower back. The mice were immunized intravenously 3 times on Days 1, 3, and 5 after tumor implantation with PBS (G1), non-DC (G2; iDC), OVA_257–264_-DC, and OVAs-DC (G3), or ALP/OVAs-DC (G4). Tumor size was measured every 3 days, and tumor volume was calculated as follows: *V = (2A × B)/2*, where A is the length of the short axis and B is the length of the long axis.

### 2.18. Statistical Analysis

Data were analyzed using Tukey’s multiple comparison test or unpaired t-tests employing GraphPad Prism 7 (2018, GraphPad, San Diego, CA, USA). The data are expressed as the mean ± standard deviation (SD). * *p* < 0.05, ** *p* < 0.01, and *** *p* < 0.001 were considered statistically significant.

## 3. Results

### 3.1. ALP Promotes the Expression of Th1-Polarizing Pro-Inflammatory Cytokines and Surface Molecules of DCs

DCs are one of the functionally specialized antigen-presenting cells that help the generation of effector and memory T cells when pulsed with antigens. Thus, the strategy of effectively inducing the maturation of DCs plays an essential role in developing new vaccines and adjuvants [24]. Matured DCs can induce the increased expression of cytokines and surface molecules involved in T cell activation and the activation of antigen-presenting ability with reduced antigen uptake [25]. Based on the characteristic of DCs, we examined whether ALP could induce DC maturation. In all DC experiments in this study, LPS was selected as the positive control for DC maturation. Before identifying the maturation-promoting effects of ALP in BMDCs, we confirmed the cytotoxic effects in ALP-treated BMDCs. As a result, no cytotoxic effects of ALP were observed at a concentration below 200 μg/mL (Figure 1A). Next, we focused on the cytokine secretion induced in ALP-treated DCs. ALP-treated DCs showed an enhanced production of pro-inflammatory cytokines, such as TNF-α and IL-12p70, in a dose-dependent manner, whereas the anti-inflammatory cytokine IL-10 was not induced (Figure 1B). These results were confirmed by intracellular cytokine staining (Figure 1C). In addition, the expression of surface molecules, such as CD80, CD86, MHC-I, and MHC-II, induced in ALP-treated DCs was higher than that in non-treated DCs (Figure 1D).

### 3.2. ALP Increases Antigen-Presenting Activity of DCs by Reducing Antigen Uptake

We analyzed the reduced endocytic capacity of DCs after ALP treatment in the presence of dextran. As shown in Figure 2A, ALP-treated DCs showed a decreased endocytic activity (CD11c^+^Dextran^+^ cells) compared to non-treated DCs. Since antigen-presenting ability is influenced by MHC-I and MHC-II expression [26], we measured the antigen-presenting ability of ALP-treated DCs using anti-Y-Ae mAb “straightly response” and anti-25-D1.16 mAb. Anti-Y-Ae mAb “straightly response” to the Ea_52–68_ peptide MHC-II and anti-25-D1.16 mAb identified the OVA_257–264_ peptide combined with the H-2Kb of MHC-I. For testing the antigen-presenting ability of ALP-treated DCs, cells (non-, LPS-, and ALP-treated DCs) were treated with the Ea_44–76_ peptide or OVA protein. After 18 h, the cells were labeled with fluorescein-conjugated anti-Y-Ae mAb (for Ea_44–76_ peptide-treated DCs) or anti-25-D1.16 (for OVA-treated DCs). Ea_52–68_ and OVA_257–264_ peptides were selected as the positive controls for MHC-I and MHC-II presentation. We found that ALP-treated DCs had an increased percentage of Ea_52–68_/MHC-II (Figure 2B) and OVA_257–264_/MHC-I (Figure 2C) complexes. These results suggest that ALP could initiate DC maturation by up-regulating Th1-polarizing cytokine production, surface molecule expression, and antigen-presenting ability. 

### 3.3. Activation of MAPK and NF-ĸB Signaling Pathways Mediates ALP-Induced DC Maturation

The MAPK and NF-ĸB pathways are the two major pathways activated by the maturation of DCs [27]. Therefore, to investigate whether ALP-stimulated DCs can induce these signals, we examined the effect of ALP treatment on the activation of the MAPK signaling pathway by analyzing the phosphorylation of MAPKs (ERK, JNK, and p38). NF-ĸB activation was also confirmed by detecting the nuclear translocation of p65 from the cytosol and the phosphorylation/degradation of IĸBα in the cytosol using Western blot analysis (Figure 3A,B) and confocal microscopy (Figure 3C). When the DCs were treated with ALP, the levels of phosphorylated ERK, JNK, and p38 were significantly increased. In addition, ALP-treated DCs promoted the phosphorylation/degradation of IĸBα in the cytosol and p65 translocation into the nucleus (Figure 3A,B). The nuclear translocation of NF-ĸB was also confirmed using confocal microscopy (Figure 3C). To confirm the involvement of MAPKs and NF-ĸB signals in the matured phenotypes induced in ALP-treated DCs, we analyzed the surface molecules of DCs in the absence and presence of specific pharmacological inhibitors of the MAPK and NF-ĸB signaling pathways. As expected, in the presence of these inhibitors, the surface molecules of ALP-treated DCs were significantly reduced (Figure 3D), suggesting that MAPK and NF-ĸB signaling are the main pathways for ALP-induced DC maturation.

### 3.4. ALP-Activated DCs Functionally Induce Naive T Cells Toward Th1, Activate CD8^+^ T Cells, and Strongly Increase CTL Activity

ALP-treated DCs were co-cultured with CFSE-labeled allogeneic CD4^+^ or CD8^+^ T cells isolated from BALB/c mice for 2 days to evaluate whether ALP-treated DCs could induce T cell proliferation. ALP-treated DCs induced the proliferation of both CD4^+^ and CD8^+^ T cells compared with non-treated DCs (Figure 4A). In the culture supernatants, we measured the expression levels of the IFN-γ, IL-2, and IL-5 cytokines produced by CD4^+^ and CD8^+^ T cells co-cultured with non-, LPS-, and ALP-treated DCs. IFN-γ and IL-2 are cytokines secreted by Th1 and activated CD8^+^ T cells, and IL-5 is a Th2-producing cytokine [28]. The results showed that CD4^+^ and CD8^+^ T cells co-cultured with ALP-treated DCs robustly secreted both IFN-γ and IL-2 compared with CD4^+^ and CD8^+^ T cells co-cultured with non-treated DCs. CD4^+^ T cells co-incubated with ALP-stimulated DCs secreted significantly lower levels of IL-5 (Figure 4B)

Mice were immunized with PBS, non-DC, OVA-DC, or ALP/OVA-DC, and each group was injected with a mixture of syngeneic CFSE^high^ and CFSE^low^. The elimination of CFSE^high^ in the splenocytes of each immunized mouse was analyzed. Increased levels of target cell lysis were observed in ALP/OVA-DC-immunized groups compared to OVA-DC-immunized groups (Figure 4C). These results indicate that ALP-treated DCs showed immunogenic potential that promoted Th1 polarization and CTL activity.

### 3.5. Systemic Administration of ALP-Stimulated OVA Peptide-Pulsed DCs Promotes the Generation of OVA-Specific Multifunctional T Cells and the Retardation of Tumor Growth

We next evaluated multifunctional CD4^+^ and CD8^+^ T cells capable of simultaneously producing multiple effector Th1 cytokines and cytotoxic markers (IFN-γ, TNF-α, IL-2, and CD107a) because these cell types are strong effectors in tumor environments [29,30,31,32,33]. Spleen cells from each immunized mouse (PBS, non-DC, OVAs-DC, or ALP/OVAs-DC) were stimulated with OVA_257–264_ and OVA_323–339_. More OVA peptide-specific multifunctional CD107^+^TNF-α^+^CD4^+^, CD107^+^IFN-γ^+^IL-2^+^TNF-α^+^CD8^+^, CD107^+^IL-2^+^TNF-α^+^CD8^+^, and CD107^+^TNF-α^+^CD8^+^ T cells were observed in the ALP/OVAs-DC group compared to OVAs-DC groups (Figure 5A). Additionally, the ALP/OVAs-DC group displayed an enhanced serum antigen-specific IgG2a response, an indicator of Th1 immunity, but not IgG1 response, an indicator of Th2 immunity (Figure 5B). 

Finally, to show the therapeutic anti-tumor potential of ALP-treated DCs, mice were injected with OVA-expressing EG.7 tumor cells. After tumor implantation, the mice were immunized (PBS, non-DC, OVAs-DC, or ALP/OVAs-DC). Interestingly, the ALP/OVAs-DC groups significantly showed retardation of E.G7 tumor growth compared to the OVAs-DC groups (Figure 5C). These results indicated that ALP-treated DCs had anti-tumor activity on the basis of the generation of antigen-specific multifunctional T cells and a Th1-supported Ab response.

## 4. Discussion

Some polysaccharides from plants, bacteria, yeasts, and synthetic sources can simulate and activate innate cell types, and regulate adaptive immunity [34]. Importantly, polysaccharides that help create a Th1-type cellular response have been considered strong adjuvant candidates in the use of vaccines against cancer, viruses, and bacteria [35,36,37]. In the current study, we found that ALP could induce the phenotypic maturation of DCs by stimulating the expression of cell surface molecules, antigen-presenting ability, and Th1-polarizing pro-inflammatory cytokines. We also found that this maturation was induced through the activation of MAPK- and NF-κB signaling pathways. ALP also participated in the initiation and regulation of adaptive immunity by shifting the polarization of naive CD4^+^ T cells toward Th1 and inducing the activation of CD8^+^ T cells, based on the mature phenotype of DCs. 

The induction of DC maturation is an absolute requirement in developing vaccines and adjuvants due to their ability to activate naive T cells. They can effectively present antigen-derived peptides to naive T cells [38,39,40]. Besides, this reaction can lead to pathogen-specific T cell immunity with an immunological memory of protective effector functions [41]. Thus, natural products that enhance DC maturation and function could be used as adjuvants for various vaccines, such as inactivated and subunit- and DC-based immunotherapeutic vaccines [42,43,44]. When the matured phenotypes and functional alterations of ALP-treated DCs were assessed, our results indicated that ALP had the potential to be an adjuvant that can increase DC maturation.

Many researchers also emphasize the need to precisely regulate T cell subtypes and phenotypes for the effective selection of an adjuvant [45,46,47]. For example, adjuvants that lead to a poor Th1- or CTL- or Th2-biased immune response are inappropriate for use as cancer vaccine adjuvants, because robust tumor-specific CD8^+^ CTLs are required for the effective anti-cancer therapies; memory Th1 cells can augment tumor-specific CTL responses, whereas Th2 cells favor tumor growth by inhibiting Th1 immunity [48,49,50]. Importantly, among the various types of activated CD8 T cells, multifunctional CD8^+^ T cells capable of simultaneously producing multiple effector Th1 cytokines and cytotoxic markers (IFN-γ, TNF-α, IL-2, and CD107a) have been associated with higher protective efficacy in cancer immunotherapy and vaccination [29,30]. Furthermore, recent studies have shown that multifunctional CD4^+^ T cells with cytotoxic activity (CD4^+^ CTL) play essential roles in anti-tumor immunity [31,32,33]. The role of these cell types has been identified in vaccine studies against various diseases [51,52]. One of the most notable features of these T cells is having higher effector functions (memory T cell expansion and cytotoxic activity) compared with monofunctional CD4^+^ and CD8^+^ T cells [29,53]. Additionally, Dahan et al. showed that IgG2a produced by B cells, which is an immunoglobulin isotype marker for Th1-type T cell immunity, is required for optimal anti-tumor activity [54]. These observations suggest that an effective adjuvant for cancer vaccines should elicit these T cell and B cell responses. Based on these trends, an examination of ALP/OVA-pulsed DC vaccination showed that ALP could induce anti-cancer activity by inducing strong CTL activity, the generation of multifunctional CD4^+^ (CD107^+^TNF-α^+^CD4^+^) and CD8^+^ (CD107^+^IFN-γ^+^IL-2^+^TNF-α^+^CD8^+^, CD107^+^IL-2^+^TNF-α^+^CD8^+^, and CD107^+^TNF-α^+^CD8^+^) T cells, and an IgG2a response. Hence, ALP has potential as an adjuvant candidate that can strongly induce antigen-specific Th1, CTL, and B cell responses.

## 5. Conclusions

In summary, the selection of effective adjuvant for various vaccine platforms requires a more profound comprehension of the immune response induced by adjuvant candidates, which is considered an integral part. Our results obtained using in vitro models showed that ALP induces robust DC maturation in a MAPK- and NF-κB-dependent manner and can induce a robust adaptive immune response by explicitly promoting the development of Th1 and activated CD8 T cells. Additionally, as demonstrated in DC-based immunotherapeutic vaccination, ALP-treated DCs induce strong immunogenicity, including CTL activity, the generation of antigen-specific multifunctional T cells, and a Th1-supported Ab response. These indicate that ALP is a potential candidate for use as an effective adjuvant that modulates strong Th1-type cell-mediated immunity and CTL responses for various vaccine platforms. 

## Figures and Tables

**Figure 1 nutrients-12-01602-f001:**
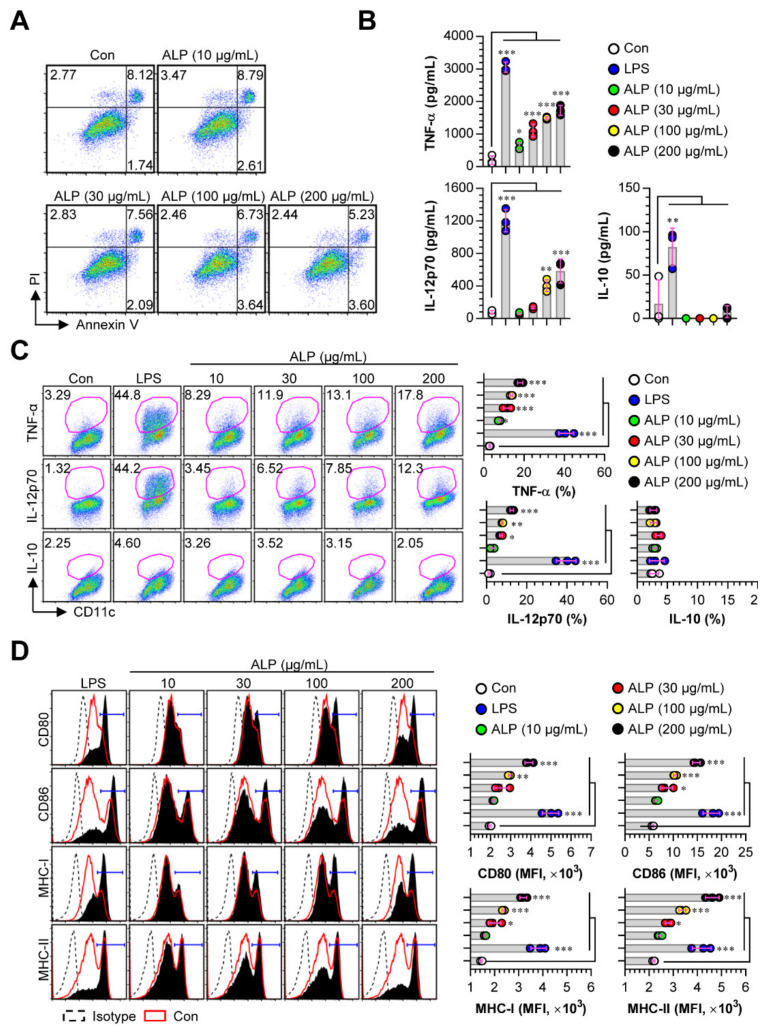
Cytotoxic activity, surface molecule expression, and cytokine production in ALP-stimulated dendritic cells (DCs). DCs were treated with *A. muricata* L. leaf polysaccharide (ALP) (10, 30, 100, and 200 μg/mL) and LPS (100 ng/mL) for 24 h. (**A**) Cell toxicity was determined using AnnexinV/PI staining (PI^+^ cells; necrosis, AnnexinV^+^PI^+^ cells; early necrosis, AnnexinV^+^ cells; apoptosis). (**B**) The culture supernatant was used to analyze the production of TNF-α, IL-12p70, and IL-10 using ELISA. (**C**) Intracellular cytokine levels of pro- (TNF-α and IL-12p70) and anti-inflammatory cytokines (IL-10) were assessed in non-, LPS-, and ALP-treated DCs. (**D**) Expression levels of CD80, CD86, MHC-I, and MHC-II in CD11c^+^-gated cells were detected by FACS. The mean fluorescence intensities of surface molecules in CD11c^+^ cells are shown in each panel. These results are indicated as the mean ± SD (*n* = 3 samples) of 3 representative experiments. * *p* < 0.05, ** *p* < 0.01, or *** *p* < 0.001. SD; standard deviation.

**Figure 2 nutrients-12-01602-f002:**
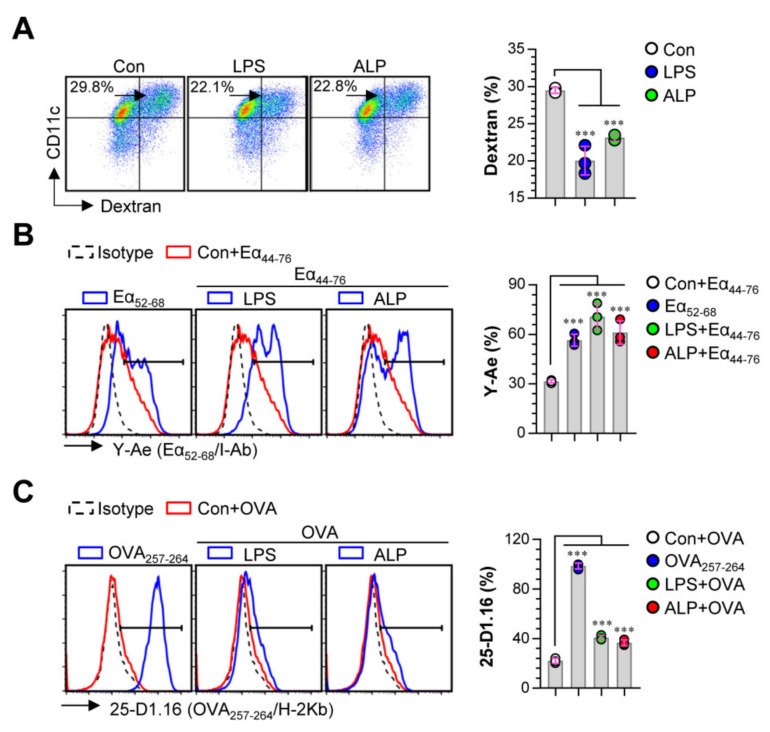
Antigen uptake and antigen-presenting ability of ALP-stimulated DCs. (**A**) Cells were stimulated with ALP (200 μg/mL) or LPS (100 ng/mL) for 18 h and cultured with dextran at 37 °C for 30 min. The cells were labeled with anti-CD11c Ab and subjected to FACS analysis to detect dextran uptake. (**B**,**C**) Non-, LPS-, and ALP-treated cells were then treated with Eα peptide (aa 44–76; 25 μg/mL) or ovalbumin (OVA) (500 μg/mL) for 24 h. After incubating, each group of cells was labeled with anti-CD11c, anti-Y-Ae, or anti-25-D1.16 mAbs for 15 min. Positive controls for antigen presentation, Eα_52–68_ (5 μg/mL) or OVA_257–264_ (5 μg/mL), were used. Histogram data and bar graphs show the expression of Eα_52–68_/I-Ab (**B**) and OVA_257–264_/H-2Kb (**C**) in the gated CD11c^+^ population. Bar graph data are shown as the mean ± SD (*n* = 3 samples) of 3 representative experiments. *** *p* < 0.001. SD; standard deviation.

**Figure 3 nutrients-12-01602-f003:**
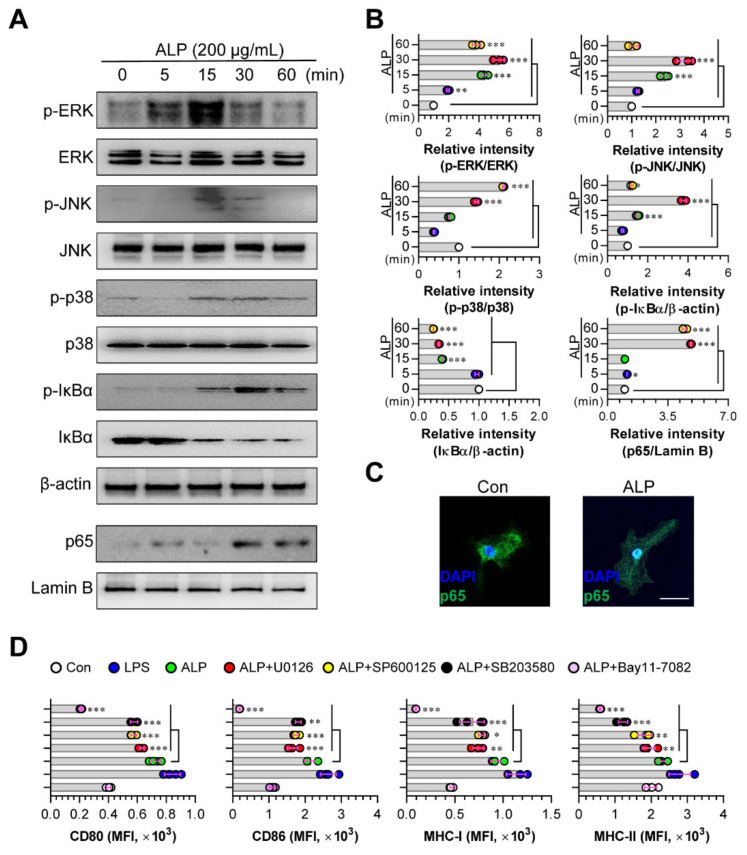
Activation of MAPK and NF-κB signaling pathways in ALP-treated DCs. (**A**) Cells were stimulated with ALP (200 μg/mL) for different times. After the incubation, the cells were lysed for protein extraction. SDS-PAGE and immunoblotting were performed to analyze the maturation signal using phosphorylated antigen-specific and total antigen-specific Abs against ERK, JNK, p38, and IĸB-α. Data are representative of 3 independent experiments. (**B**) Western blotting data were analyzed using the Image J software to compare the phosphorylated and total forms. (**C**) The effect of ALP on the nuclear translocation of p65 from the cytosol. Cells were stimulated with ALP (200 μg/mL) for 1 h and stained with DAPI and Alexa 488-conjugated anti-NF-κB (Scale bar: 10 μm). (**D**) DCs pre-treated with the MAPK and NF-κB signaling pathway inhibitors, U0126 (ERK), SP600125 (JNK), SB203580 (p38), and Bay11-7082 (NF-κB) were incubated with ALP (200 μg/mL) for 18 h. After incubation, the cells were assessed using FACS to detect the expression levels of CD80, CD86, MHC-I, and MHC-II. The results are indicated as mean ± SD (*n* = 4 samples), representative of 3 experiments. * *p* < 0.05, ** *p* < 0.01, or *** *p* < 0.001. SD; standard deviation.

**Figure 4 nutrients-12-01602-f004:**
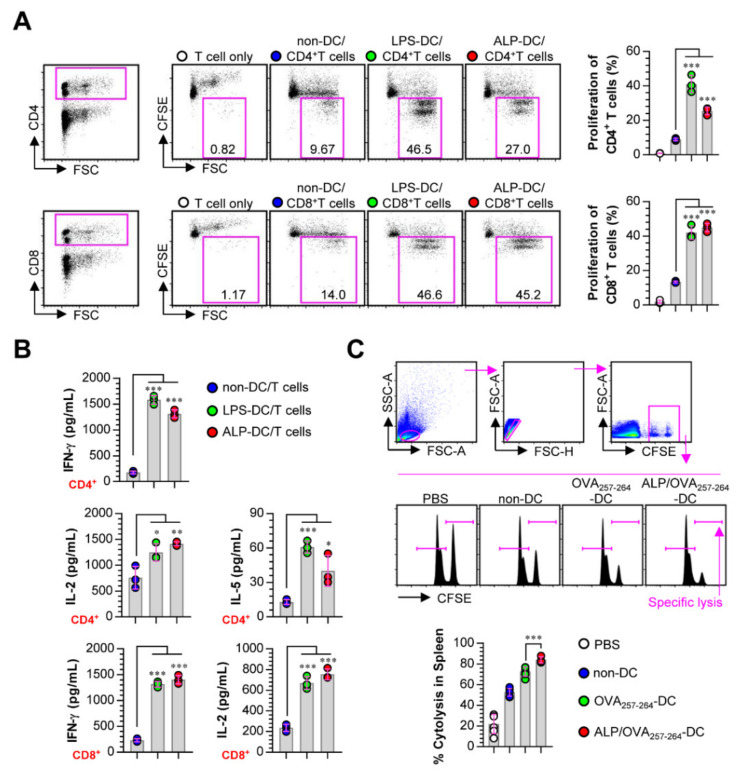
ALP-treated DCs initiate T cell proliferation and the Th1 response and enhance cytotoxic T lymphocyte (CTL) activity. (**A**) ALP (200 μg/mL)-treated DCs were co-incubated with carboxyfluorescein succinimidyl ester (CFSE)-stained BALB/c background CD4^+^ and CD8^+^ T cells at a ratio of 0.2:1 (DCs and T cells, respectively). After 2 days of co-culture, the proliferation of CD4^+^ or CD8^+^ T cells was assessed by FACS. The data shown are representative of a set of 3 independent experiments. (**B**) Culture media were used for the analysis of the expression levels of the indicated cytokines using ELISA. (**C**) In each immunized mouse, CTL activity was measured with CFSE^high^ OVA_257–264_-loaded and CFSE^low^ non-loaded splenocytes. Data show the mean ± SD of the percentages of particular lethality, representing 2 independent experiments, *** *p* < 0.001. SD; standard deviation.

**Figure 5 nutrients-12-01602-f005:**
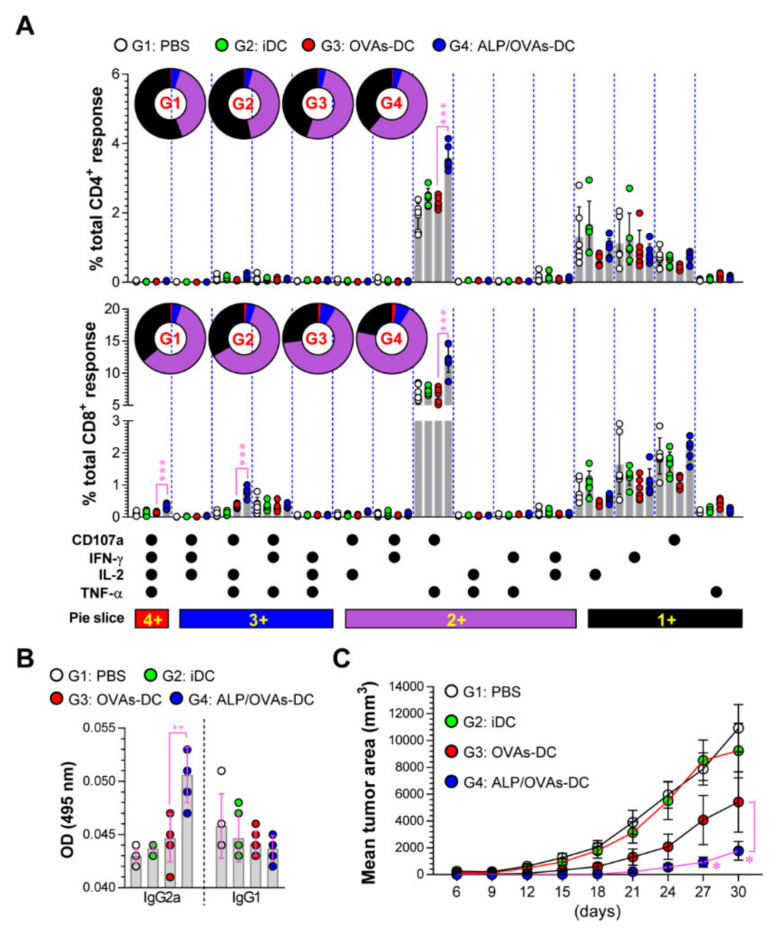
Immunization with ALP-treated DCs induces a robust anti-tumor effect and immunity. (**A**) Two weeks after the final immunization, single cells obtained from PBS-, iDC-, OVAs-DC-, and ALP/OVAs-DC-immunized mice were stimulated ex vivo with OVA peptides for 6 h. The percentage of OVA-specific CD3^+^CD4^+^ and CD3^+^CD8^+^ T cells producing CD107a, IFN-γ, IL-2, or TNF-α was analyzed in accord with the gating strategy described in Appendix A (Appendix A). Pie charts indicate the mean frequencies of CD4^+^ and CD8^+^ T cells co-expressing IFN-γ, TNF-α, IL-2, and CD107a. (**B**) Serum samples were analyzed using ELISA for OVA_323–339_-specific IgG2a and IgG1 antibodies in each immunized mouse. The mean ± SD (*n* = 6 mice/group) shown are representative of 2 independent experiments. (**C**) Tumor volumes in mice (*n* = 10 mice/group) that received PBS (G1: white solid dot), iDC (G2: green solid dot), OVAs-DC (G3: red solid dot), and ALP/OVA-DCs (G4: blue solid dot) vs. E.G7. Tumor growth was identified by calculating the diameter of the tumor every 3 days for 30 days. The mean ± SD shown are representative of 2 independent experiments. SD; standard deviation.

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
