# Peer review of "Annona muricata L.-Derived Polysaccharides as a Potential Adjuvant to a Dendritic Cell-Based Vaccine in a Thymoma-Bearing Model"

_nutrients, 2020, doi:10.3390/nu12061602_

Round 1
Reviewer 1 Report
This manuscript described the potential usage of plant extracted ALP as adjuvant for DC vaccine. The flow of the experimental design is logical and most of the data are adequately presented. The only concern is OVA antigen specific tumor/cells were exploited in this MS, which respond well to OVA peptide treatment, with or without using DC as delivery vehicle in conjunction to proper adjuvant, therefore data are not adequate to support the claim. In addition, faults were found as follow:
Line 255-256 incomplete sentence “DCs were Table 10. …for 24h”
Line238 confusing title of section 3.1? Contents in this section is not relevant to the title. Physical characterization (e.g. purity, activity, half-life) of extracted ALP is lacking.
Figure 5A Discrepancy in data presentation. For instance, in G2 group (blue solid dot) ~0.8% of CD4+ T cells was CD107a+ and less than 0.2% was TNFa, however ~3.8% of them was double positive for CD107a and TNFa, how this happen? I found this figure is very confusing, and the figure legend is misleading, as G2 iDC group induced the highest cytokine production among different cytokine combination?
Line 357 “…enhanced serum antigen-specific IgG2c response…” typo
Reviewer 2 Report
Dear Authors,
I was pleased to review the assigned manuscript. Overall, a proposed manuscript is well written. The study does have few very minor issues (generally, editorial errors or oversights). Detailed comments are included in the PDF file.
All best,

Reviewer 3 Report
The manuscript showed Annona muricata leaves polysaccharides (ALP) as a potential adjuvant to dendritic cell-based vaccine in a thymoma-bearing model. Introduction and methods are written well and supported with references. Results are well determined through graphical representation. However, there are few minor comments need to be addressed:
Page 1; Line 28: Need to write significant figure or percentage
Page 1; Line 30: Need to write significant figure or percentage
Page 2; Line 92: What was the passage of cells?

Round 2
Reviewer 1 Report
I was pleased to review the reversed version of the assigned manuscript again. The questions I had previously were well addressed by the authors.